# Neuroprotection of the Perinatal Brain by Early Information of Cerebral Oxygenation and Perfusion Patterns

**DOI:** 10.3390/ijms22105389

**Published:** 2021-05-20

**Authors:** Filipe Gonçalves Costa, Naser Hakimi, Frank Van Bel

**Affiliations:** 1Department of Neonatology, University Medical Center Utrecht, 3584 EA Utrecht, The Netherlands; fgervas2@umcutrecht.nl (F.G.C.); naser@artinis.com (N.H.); 2Artinis Medical Systems, B.V., 6662 PW Elst, The Netherlands

**Keywords:** cerebral oxygenation monitoring, near infrared spectroscopy, neuroprotection, neuromonitoring, cerebral perfusion, cerebral autoregulation

## Abstract

Abnormal patterns of cerebral perfusion/oxygenation are associated with neuronal damage. In preterm neonates, hypoxemia, hypo-/hypercapnia and lack of cerebral autoregulation are related to peri-intraventricular hemorrhages and white matter injury. Reperfusion damage after perinatal hypoxic ischemia in term neonates seems related with cerebral hyperoxygenation. Since biological tissue is transparent for near infrared (NIR) light, NIR-spectroscopy (NIRS) is a noninvasive bedside tool to monitor brain oxygenation and perfusion. This review focuses on early assessment and guiding abnormal cerebral oxygenation/perfusion patterns to possibly reduce brain injury. In term infants, early patterns of brain oxygenation helps to decide whether or not therapy (hypothermia) and add-on therapies should be considered. Further NIRS-related technical advances such as the use of (functional) NIRS allowing simultaneous estimation and integrating of heart rate, respiration rate and monitoring cerebral autoregulation will be discussed.

## 1. Introduction

Reduction and prevention of perinatal brain damage focus, among other measures, mainly on stress reduction (preterm neonate), moderate hypothermia (term neonate) and pharmacological (add-on) therapy [1,2,3]. However, abnormal oxygenation and/or perfusion patterns often precede brain damage. The occurrence and extension of peri-intraventricular hemorrhages (PIVH) in very and extremely preterm infants (born between 28–32 weeks gestational age and below 28 weeks, respectively), whereas profound and persistent hypoxemia or hyperoxemia often precedes white matter injury (WMI) in this vulnerable group of neonates [4,5]. Likewise, in term neonates experiencing perinatal hypoxia-ischemia or birth asphyxia, (early) cerebral hyperoxygenation, loss of cerebral autoregulation and subsequent perfusion disturbances are related with additional brain damage and adverse 2-year outcome, or even death [6,7,8]. Early identification of erroneous cerebral oxygenation and/or perfusion patterns may therefore be important for timely corrective interventions, if possible, in order to prevent or at least reduce damage to the developing brain.

Biological tissue is transparent for the near infrared light part of the spectrum (700–1000 nm) and its subsequent absorption by oxygenated and deoxygenated hemoglobin makes (continuous-wave) near-infrared spectroscopy (NIRS) an ideal bedside tool to noninvasively monitor substantial changes in oxygenation and perfusion of the developing brain [9,10,11,12]. It actually measures the regional cerebral saturation (rScO_2_) in a mixture of arterial, capillary and venous blood (normally in a ratio of ~25~5~75%, respectively) within the reach of the NIRS sensor attached to the skin of the head [9]. Typically, a banana-shaped area between emitting and receiving optodes is measured. The maximal penetration depth of the near infrared light is between 2 and 3 cm, at least when the optodes distance is at least 4 cm [13]. It is expected that white and grey matter are both “insonated”, at least in the preterm infants. This may be less straight forward when dealing with full-term infants. Oxygenation of the different brain regions seem to be quite homogeneous and did not substantially differ as investigated by us [14], suggesting that a single sensor placement is reliable to determine global rScO_2_. It is important to state that the clinically used NIRS devices monitor “trends” or changes in rScO_2_ rather than absolute values. Earlier studies showed that the precision of the commercially available devices is about 5.2% (ability to measure the same value after repeated measurement attempts) [9,10,15]. Moreover, there are sometimes quite significant differences in readings of rScO_2_ by the various available devices based on different basic assumptions or algorithms [10,16]. It is widely accepted, though, that “normal” values of rScO_2_ in neonates are between 55–85% [17,18]. In several experimental and clinical studies, it has been shown that sustained episodes of rScO_2_ values below 40–45% compromise oxygen metabolism of the developing brain and may cause neuronal damage [19,20,21,22].

Despite the reservations with respect to the exactness of NIRS-determined cerebral oxygenation we are of the opinion that “trend monitoring” of the oxygenation of the immature brain at the bedside in this noninvasive manner can be relevant for clinical care. The early recognition of (persistent) abnormal patterns of rScO_2_ values and NIRS-derived cerebral oxygen extraction may reflect derailment of cerebral oxygenation and perfusion [15,23,24] and energetic and quick correction, if possible, may protect the immature brain.

In the next sections we will concentrate on the possible benefits for clinical care of early evaluation of the NIRS-monitored oxygen saturation signal (pattern recognition) in extremely preterm infants [15] and in term infants who experienced perinatal hypoxia and ischemia [8]. Further refinement such as integration with other monitoring parameters such as heart rate, respiratory rate and amplitude-integrated EEG (aEEG) will be briefly discussed, as well as the benefits of integration of the NIRS signal with blood pressure to assess (the lack of) cerebral autoregulation.

## 2. Mechanisms of Brain Injury in the Preterm Infant and Cerebral Oxygenation

In extremely preterm neonates, PIVH is one of the most important reasons for adverse (motor) developmental outcome and death, which has an incidence up to 45% in this population [25,26]. Mostly the bleedings are limited to the germinal matrix or with an additional moderate extension into the lateral cerebral ventricles (grade I and II, respectively, according to the Papile classification [27]). A minority evolves to larger PIVHs, grade III (large amount of blood in the cerebral ventricles which are dilated) or even extension of the blood into the brain parenchyma around the cerebral ventricles (grade IV or venous infarction). The etiology of PIVH is multifactorial and PIVHs developed perinatally and/or extended within 12 h after birth seem strongly associated with pro-inflammatory cytokines and formation of oxidative species [28,29]. However, most PIVHs develop and extend during the first 3 days of life or even later. The combination of vascular immaturity in the microcirculation of the germinal matrix [30] and often substantial increases (and fluctuations) in brain perfusion due to the respiratory distress syndrome (IRDS)-induced acutely increases in pCO_2_ and/or hypoxia [31,32], setting the brain for the development of PIVHs and their extension to more severe stages (i.e., grade III and IV) [28,33]. Important in this respect is that intravascular CO_2_ diffuses very quickly to the perivascular space of the cerebral resistance vasculature (contrary to bicarbonate: the blood–brain barrier is relatively impermeable to bicarbonate [34]), where it lowers the perivascular pH, inducing acute vasodilation [32]. The fragile autoregulatory ability of the cerebral vascular bed in these extremely immature neonates is further provoked by the vasodilating effect of hypercarbia. This could result in a blood pressure-passive brain circulation contributing to fluctuations in the cerebral perfusion, possible episodes of hypoxia during hypotension and to extensions of already existing bleedings [35,36].

It may be clear that prevention or early recognition and down tuning of cerebral hyper(hypo-)perfusion/-oxygenation and/or a fluctuating pattern of the cerebral blood flow potentially reduce or even prevent the occurrence and extension of PIVHs. In this respect it is especially important to aim for a stable arterial carbon dioxide level within normal limits (i.e., 35–50 mmHg). Monitoring the pattern of cerebral oxygenation using NIRS-determined rScO_2_ as early after birth as possible, preferably already on the resuscitation table, at least up to postnatal day 4 in these vulnerable group of neonates can alert the clinician at an early point in time for hypercapnia-induced hyperoxygenation/hyperperfusion [25,35] and hypoxia- or hypocapnia-induced underoxygenation/hypoperfusion [32] of the immature brain. A very recent study from Tamura et al. [37] showed that hypercapnia in the delivery room was associated with the occurrence of all grades of PIVH in a large cohort of preterm infants. Alderliesten et al. [35] showed supranormal rScO_2_ values [17] during 24 h preceding PIVH in a case–control study of 650 preterm infants below 32 gestational weeks. In the same study it was shown that the combination of hyperoxygenation (hyperperfusion) and loss of autoregulation preceded the extension of PIVHs to a higher grade. It was also reported that inotropes were related with loss of cerebral autoregulation (defined as a moving correlation >0.5 between rScO_2_ and mean arterial pressure). Other potentially more precise mathematical models to describe presence or absence of cerebral autoregulation such as coherence and transfer function gain are intensively investigated on their clinical relevance [38], which will be discussed in more detail in another section of this paper. A follow-up study at 15 and 24 months corrected age of a cohort of 734 preterm infants showed an association between low rScO_2_ values and adverse neurodevelopmental outcome at 2 years of age [39]. A recent prospective study in 185 extremely preterm neonates using NIRS-monitored rScO_2_ showed that prolonged cerebral desaturation was associated with the occurrence of any grade of PIVH [40]. So, early intervention to quickly normalize the arterial carbon dioxide, cerebral oxygen saturation levels, and prevention of large fluctuations in cerebral perfusion due to lack of autoregulatory ability of the cerebral vascular bed (“hands off” policy and/or sedation) will probably lower the PIVH incidence and its severity [41]. Up to now, only one prospective randomized intervention study in which in 166 preterm infants born after less than 28 weeks cerebral oxygenation was monitored with NIRS (study group) or not (control group) aiming to reduce episodes of hypo- or hyperoxia during the first 3 days of life [42]. It was reported that monitored infants, contrary to those infants whose monitors were blinded, lowered the burden of hypoxia and/or hyperoxia and showed a more stable cerebral oxygen saturation which was mostly within normal limits (55–85%) [17]. This decreased the incidence of particularly severe PIVHs (grade III and IV) [43].

Although the incidence of cystic periventricular leukomalacia has greatly decreased in modern neonatal intensive care units, diffuse white matter injury (dWMI) occurs in about 50% of the extremely and very preterm infants according to MRI studies [44,45]. It has been suggested that an aborted maturation of the very vulnerable precursors of oligodendrocytes and by that arrest of myelination is responsible for the dWMI [46] and a very important reason for a suboptimal neurodevelopmental outcome of this patient group [47]. dWMI is often related to maternal and/or fetal infection or fetal hypoxia leading to pro-inflammatory cytokines and increased activity of microglia disturbing the development of oligodendrocyte precursors [48]. However, postnatal issues and in particular hypoxia with or without ischemia are also strongly related to dWMI. A relatively important role postnatally plays acute hypocapnia (and to a lesser extend hyperoxia) leading to instantaneous vasoconstriction of the cerebral vascular bed with consequent hypoperfusion, and thus lower oxygen delivery [5,32]. Especially abnormally low rScO_2_ values in a stable but artificially ventilated infant is suspicious for a ventilation-induced hypocapnia leading to ischemic-hypoxic damage of the immature brain [49]. The combination of NIRS-monitored cerebral oxygenation, arterial oxygen saturation, and end-tidal CO_2_ monitoring, increasingly used in standard clinical care in newborn intensive care units, might be a strong preventive policy against hypocapnia-induced WMI.

Some caution is also justified in case of a longstanding hemodynamically Patent Ductus Arteriosus (hsPDA), especially at present where an expectative policy with regard to (surgical) closure of the duct becomes mainstream [50,51]. A hsPDA gives rise to a ductal steal of blood originally destined for the brain with potential underperfusion/oxygenation of the developing immature brain and may not match metabolic demands [52]. In a study with a relatively large population of infants with hsPDA, we found abnormally low values of rScO_2_, often below 45% (so well below the lower limit of “normal” values of 55% [17]). Persistent values lower than 40–45% are reported to be related with brain damage [19,53]. We indeed found an association between low cerebral oxygenation and duration of a hsPDA on the one hand and a lower volume of especially the cerebellum, a brain region with a remarkably high metabolism, on the other hand [53,54,55]. This finding confirmed earlier studies which reported negative effects of hsPDA on cerebellar growth in these immature neonates [22,56]. This suggests that longstanding hsPDAs may not always be as innocent as earlier believed. In our NICU all infants with signs of an hsPDA the cerebral oxygenation will be monitored for reasons as explained above.

The definition of “hypotension” of prematurity, mostly defined as a mean arterial pressure lower than the gestational age in weeks, gives way to overtreatment with positive inotropes. Several studies report the absence of signs of organ system dysfunction, metabolic acidosis or NIRS-monitored under-oxygenation of the brain during these “hypotensive” periods [35,57]. Moreover, long-term developmental outcome was not different compared to infants without hypotension [57,58]. At present our policy toward “hypotension” is expectative when NIRS-determined oxygenation of the brain is within normal limits with no signs of systemic under-perfusion or acidosis.

Other issues in which NIRS-monitored cerebral oxygenation can help to prevent injury to the immature brain are anemia and hypoglycemia: anemia with hemoglobin values below 6 mmol/L is reported to affect proper oxygenation of the preterm brain [59], whereas profound hypoglycemia can be recognized by spontaneously increasing levels of cerebral oxygenation due to depletion of glucose leading to a hampering of metabolism and hence oxygen utilization [60].

## 3. Mechanisms of Brain Injury of the Term Infant and Cerebral Oxygenation

Hypoxic-ischemic encephalopathy (HIE) due to perinatal asphyxia is a major cause of adverse neurodevelopmental outcome and neonatal mortality [61]. Although neuronal damage occurs during the actual (mostly fetal) hypoxic episode(s) due to excessive formation of excitatory neurotransmitters, profound calcium influx into the neuronal cells leading to acute oxidative damage to the cell membrane, a substantial part of birth asphyxia-induced brain injury occurs during the first 48 to 96 h of life and can even last for weeks and starts upon reperfusion and reoxygenation at birth [62]. Phosphorus nuclear magnetic resonance spectroscopy showed that during the first hours after reperfusion/reoxygenation upon birth cerebral oxidative metabolism is apparently normal, but secondary cerebral energy failure started after 10 to 12 h of age which seems to be maximal between 24 and 72 h of age leading to substantial additional brain injury [22,62]. Reoxygenation-induced production of free radicals, nitric monoxide and other toxic compounds give not only rise to acute damage to neuronal cells but also to activation of pro-inflammatory cytokine production leading to secondary cerebral energy failure and a decrease in trophic factors [63]. This delayed damage due to secondary energy failure, provides us with a “therapeutic” window in which reduction of brain damage can be achieved by preventing and/or scavenging free radical and nitrosative stress, important toxic substances contributing to secondary energy failure and concomitant brain damage. Until now, moderate hypothermia of the body down to 33.5 °C for 72 h is the only established therapy reducing reperfusion damage after birth asphyxia. However, add-on therapy with (combinations of) pharmacologic neuroprotective compounds are currently investigated: It seems very likely that the combination of hypothermia with pharmacological compounds contribute to a further reduction of birth asphyxia related brain damage [63].

For an optimal reduction or even prevention of birth asphyxia-induced reoxygenation–reperfusion damage of the developing brain, it is crucial to select infants that will benefit from postnatal treatment. This selection needs to be done as early as possible in this “therapeutic” window (alleged span of time first 6 h of life).

Abnormally high values of rScO_2_ (>80%) during the first 72 to 96 h of life in perinatally moderately and severely asphyxiated neonates, probably indicating less utilization of oxygen due to secondary energy failure and ongoing brain damage, are related with adverse long-term neurodevelopmental outcome with a high predictive value at as early as 3 h of life [8,64,65]. The pattern of rScO_2_ can, therefore, be an attractive noninvasive means to determine whether or not (add-on) neuroprotective therapy is necessary to reduce brain damage: although not properly investigated yet, a normalization of rScO_2_ values within normal limits upon (add-on) therapy may indicate a beneficial effect of treatment.

Since neuroimaging techniques such as cranial ultrasound, nowadays standard implemented in clinical care in the newborn intensive care unit, and (advanced) MRI of the neonatal brain improved substantially in the recent years, perinatal arterial ischemic stroke (PAIS) appeared to occur more often than previously reported and with an incidence of at least in 1:2300 newborns it is no longer a rare complication [66]. PAIS presents itself mostly in the first days after birth with seizures and can lead to cerebral paresis and to cognitive and motor impairments [67,68]. Neuroprotective/neuroregenerative therapeutic options were not existent up to recently and treatment was mostly limited to anti-epileptic therapy. However, nowadays potentially beneficial early neuroprotective intervention strategies are reported such as the use of neurotrophic substances such as erythropoietin and mesenchymal stem cells to boost neuro-regeneration [69,70].

Neuromonitoring, i.e., the combination of NIRS-monitored cerebral oxygenation and amplitude-integrated encephalography (aEEG), has an important prognostic value in PAIS [71]. Moreover, the early pattern of rScO_2_ and electrical background pattern and sleep-wake cycling of the aEEG are probably useful in identifying neonates with PAIS who may benefit from neuroprotective interventions and for assessment of beneficial effects of these interventions [71]. rScO_2_ is substantially higher in the affected brain half as compared to the contralateral side, probably indicating luxury perfusion in the affected hemisphere [72,73]. The magnitude and duration of this difference was related with neurodevelopmental outcome [71]. Likewise, time to normal aEEG-derived electrical background (continuous normal voltage) of the affected brain half was longer as compared to the healthy brain half. The time to normalization of the electrical background pattern of the affected side was also related to neurodevelopmental outcome [71]. It is conceivable that neuroprotective interventions can shorten the duration of these asymmetries indicating a therapeutic effect. This assumption, however, remains to be investigated.

Table 1 summarizes the association between the nature of preterm and term acquired brain injuries on the one hand and the pattern of cerebral oxygenation on the other.

## 4. Future Perspectives of Multimodal Cerebral Oxygenation Monitoring: What Can It Add to Neuroprotection of the Perinatal Brain in the Clinical Setting?

NIRS-derived monitoring of the cerebral oxygenation pattern in the preterm neonate and to a lesser extent also in the term neonate, has undeniable neuroprotective potential as showed above. However, applied research of NIRS-derived surveillance of oxygenation, cerebral perfusion and cerebral autoregulation is progressing quickly with the ultimate goal to improve clinical care.

In this perspective we want to discuss two important developments in perinatal NIRS research which may lead to clinical application in the near future: proper assessment of cerebral hemodynamics regulation which aims to couple systemic (hemodynamic and oxygenation) variables to NIRS signals; a more detailed analysis of NIRS-derived intensity signals for the surge to other usable physiological variables and artifacts detection is reviewed in Section 5 and Section 6 below.

## 5. Assessment of Cerebral Hemodynamics Regulation Using NIRS Signals

In regard to cerebral hemodynamics regulation, one must take into account the mechanism of Cerebral Vascular Autoregulation (CVAR)—or pressure-flow reactivity. This property of the brain has the end goal of maintaining the cerebral blood flow at a constant level, regardless of the systemic blood pressure (BP)—but still within a limited range of BP values [74,75]. In the case of the BP going too low (or too high), the CVAR mechanism, through the arterioles, can no longer regulate the transmural pressure and the blood flow in the brain will, respectively, decrease or increase [76] (see Figure 1).

Term infants have functional autoregulation and vasoreactivity, but preterm infants are born with an underdeveloped cerebral vascular system [74,77]. This is why the study of CVAR becomes relevant, when one takes into consideration the fact that (very) preterm infants are exposed to a higher risk of hemodynamic instability—and, therefore, brain damage—leading to PIVH or white matter injury [78] as also stated above. For this reason, online assessment and monitoring of the brain tissue oxygenation, combined with the state of the autoregulatory mechanism, is of utmost importance in the frontline of preventing brain injury and further neurodevelopmental disorders [39,79].

By evaluating the relation between BP and Cerebral Blood Flow (CBF) in a continuous way, it is possible to assess the mechanism of CVAR [75]. Since there is not a direct and non-invasive way to measure CBF, and assuming a constant arterial oxygen saturation (SaO_2_), the NIRS signal can be used as a surrogate for CBF changes, providing information on the oxygen saturation of the brain (rScO_2_) [39,80,81]. By observing the rScO_2_ signal it becomes possible to detect sudden changes and eventually relate them to variations in BP, therefore identifying the presence or absence of the dynamics of autoregulation and providing us the possibility to interfere as already suggesting above [74].

The first steps into this analysis focus on the pre-processing stage, including filtering of the data, artifact removal and SaO_2_ correction [39]. Once this step is performed, there are two main approaches possible: analysis in the time domain or in the frequency domain. Time domain analysis calculates the correlation between the BP and rScO_2_ over short continuous segments, which can then be averages and distributed according to the corresponding BP values, thus having one correlation value per “bin” of blood pressure value [80,82,83,84]. Frequency domain analysis also explores the relation between these two signals, but with a focus on specific frequency bands [39]. This might come as an advantage when one considers that CVAR may be composed of responses with different time lags, therefore taking into account the delay between changes in BP and rScO_2_ [39]. One of these methods focuses on the Transfer Function (TF), analyzing the power spectral density (PSD) of the signals, which is a representation of the presence of each different frequency in the segments considered. From this, one can assess the TF gain, which indicates the impact that a change in the input signal (the BP) had in the output signal (the rScO_2_), and therefore assessing if CVAR was indeed present or not [39,77]. Another method of analysis is Coherence: this function measures the linearity of the two signals considered (BP and rScO_2_) in the frequency domain, indicating if there is a linear relationship between the input and the output of the system in focus [39,76,77,80,84].

In all the methods described above, one must take into account the length of the epochs considered for analysis, as shorter epochs might produce noisier values—due to a smaller number of sub windows in the analysis—but epochs that are too long may indicate the presence of more nonstationary segments than the ones regarding CVAR, and, therefore, producing lower values and hindering the results [84]. Another factor from the clinical care point of view is that it is important to have in mind that these methods require signals that are continuous and stable enough to provide trustworthy results [85].

We, therefore, must always aim for a stable CVAR mechanism, even though there is still a lack of agreement as to what the best method for analysis is, as well as a proper online monitoring method at the bedside.

Despite the methods presented above, a global consensus is still lacking, as the ideal (online) system should be easily reproducible regardless of the population in focus and the preterm age. Ongoing research of CVAR assessment should also focus on the demonstration of a globally accepted tool or method that monitors CBF and rScO_2_, confirming the relation of the changes in those signals with neurodevelopmental outcome [76].

## 6. NIRS-Derived Intensity Signals and the Importance of Signal Quality Assessment

Using the NIRS technique, different information can be acquired from the intensity of the light measured by the receivers. As stated above cerebral oxygenation (rScO_2_) is the most common information acquired when using NIRS in clinical applications. However, the NIRS-derived intensity signals provide, apart from estimating the cerebral oxygenation, some useful additional functional and physiological information. The intensity signals can be converted into concentration changes in oxygenated hemoglobin (O_2_Hb) and deoxygenated hemoglobin (HHb) [86], which can be used to determine brain hemodynamic response in different brain regions [86,87,88]. The intensity and concentration signals contain some physiological information including cardiac pulsation, respiration, and vasomotor action [89,90,91,92]. Figure 2 shows an example of the O_2_Hb and HHb signals, in which the heartbeats are visibly seen with approximately 0.4-s time intervals, together with the rScO_2_ measured by Artinis cerebral oximetry device (TOM, Artinis Medical Systems B.V., Elst, The Netherlands).

### 6.1. NIRS-Derived Heartbeat Pattern

The most prominent physiological information in NIRS signals are the pulsatile fluctuations caused by heartbeats and will be discussed here. This Information in the NIRS signals allows for the extraction of heart rate (HR). Several studies have shown the compatibility of the HR derived from NIRS with the HR extracted from electrocardiography and photoplethysmography [93,94,95]. Moreover, the HR derived from NIRS has shown to be a richer source of information than HR derived from PPG or ECG under physical and mental stressful conditions [92,94,95,96]. Deriving HR from NIRS signals is highly useful in neonates as it can eliminate the need to use extra electrode for recording this information [97]. Eliminating extra electrodes can relieve neonates from physical stresses induced by electrode placement and potentially reduces injury risks [98].

The HR and brain hemodynamic information in the NIRS signals can be fused in different applications where both brain and heart functions are altered. For instance, they can be combined to accurately assess (physical) stress [92,94,97]. Furthermore, extracting the HR information from NIRS can reduce the ambiguity of the hemodynamic response model [96,99,100,101]. The extracted HR information can be used as a known physiological term in a part of the model representing different physiological components in the NIRS signal. This can ameliorate the hemodynamic response model and enhance the accuracy of brain hemodynamic response estimation in the clinical setting.

### 6.2. Signal Quality Assessment

In NIRS techniques, the NIR light travels through different tissue layers including intra- and extracranial layers before it strikes the brain. Some factors in the extracranial layer directly affect the optical pathway of the transmitted light. For example, scalp thickness, skin properties, and the presence and color of hair can disrupt the light passing through the head [102,103]. As a consequence, the coupling between NIRS sensors and scalp is decreased, and the NIRS signal quality is then deteriorated [104]. This may result in wrong interpretations and consequent false findings in the analyses. Therefore, assessing the quality of the NIRS signals before analyzing the signals is important that can increase the reliability of the statistical analysis and decrease misinterpretations.

To assess NIRS signal quality, the presence of a strong cardiac component has been often used as the main indicator of a reliable sensor-scalp coupling [105,106,107,108]. A reliable sensor-scalp coupling guarantees that NIR light travels through both intra- and extracranial layers. As a result, the received light (NIRS signals) contains brain hemodynamic and physiological information from both layers. Therefore, the presence of heartbeat, the most prominent systemic information in NIRS signals, indicates that sensors have been well coupled to the scalp and that enough light strikes the brain and is backscattered to the receiver. This knowledge will greatly improve the reliability of the NIRS signal and hence its clinical use.

## 7. Conclusions

Wide-scale utilization of NIRS-derived monitoring of cerebral oxygenation based on timely recognition of an abnormal pattern of cerebral oxygenation seems a relatively straightforward means for prevention and/or reduction of brain injury in very and extremely preterm neonates (i.e., PIVH and dWMI) and to a lesser extent also in the term asphyxiated neonate. Despite this it is very important that the precision of NIRS monitored cerebral oxygen saturation further increases. Equally important is the extended research of NIRS signals to reliably monitor CVAR and the use of the NIRS signal (quality and diversity) to monitor also hemodynamic functions, which contributes to a more informed decision on whether or not to handle the infant at a certain time. These developments may further contribute to the prevention of perinatal brain damage. International collaboration and unified definitions are important to increase utilization of NIRS as a potentially neuroprotective strategy.

## Figures and Tables

**Figure 1 ijms-22-05389-f001:**
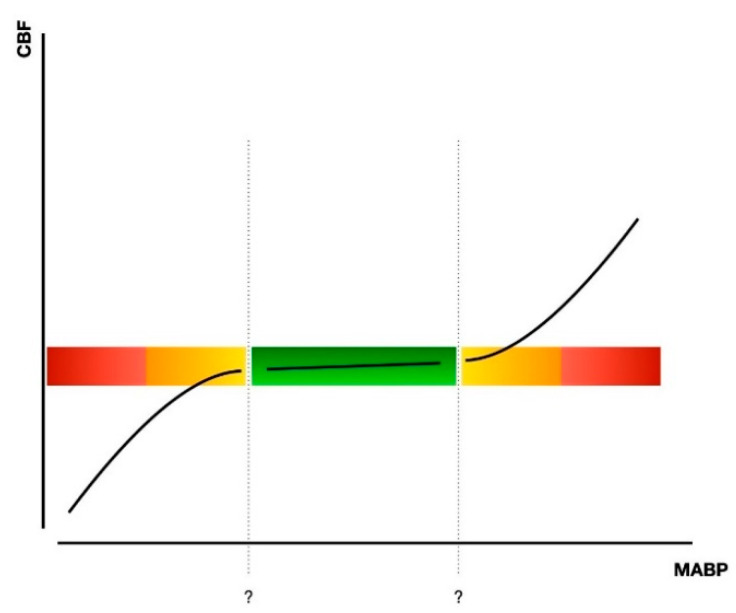
Autoregulatory curve of CBF variations according to mean arterial BP (MABP). In green, the ideal/healthy autoregulatory plateau, and in read/orange the extreme, and therefore dangerous areas of the plot.

**Figure 2 ijms-22-05389-f002:**
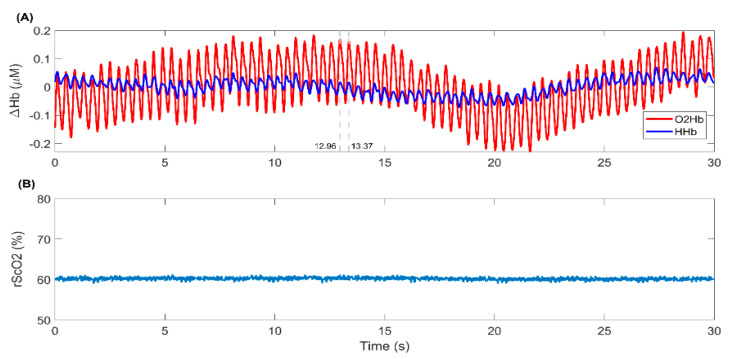
An example of the NIRS signals recorded by the TOM device (Artinis Medical Systems B.V., Elst, The Netherlands). (**A**) Concentration changes in oxygenated hemoglobin (O_2_Hb) and deoxygenated hemoglobin (HHb) in red and blue, respectively. These signals have been detrended to have a similar baseline. The heartbeats are visible in both signals as pulsatile changes with approximately 0.4-s time interval. (**B**) The cerebral tissue oxygenation, rScO_2_, computed by the TOM device in percentage. The signals are sampled at 100 Hz.

**Table 1 ijms-22-05389-t001:** Summary of associations between pathological substrate and pattern of cerebral oxygenation.

Development/Extension of PIVH
- Hypercarbia-induced cerebral vasodilation: (abnormally) high rScO_2_
- Lack of CAR: Blood pressure passive fluctuating pattern of rScO_2_
**(Cystic) Diffuse White Matter Injury**
- Hypocarbia-induced cerebral vasoconstriction: (abnormally) Low rScO_2_
- Hyperoxia-induced formation of cytokines/free rad: Low-to-normal rScO_2_
- hsPDA-related ductal steal of brain perfusion: prolonged episodes of low (<45%) rScO_2_
- Anemia-induced hypoxemia of the preterm brain: Low rScO_2_
- Hypoglycemia-related disturbance of glucose metabolism: increasing rScO_2_ values
**Perinatal Ischemia-Hypoxia (Birth Asphyxia)**
- Secondary energy failure-induced: (abnormally) high rScO_2_ (from birth up to 72–96 h of age)
**Perinatal Arterial Ischemic Stroke**
- Luxury perfusion-induced: higher rScO_2_ in the ipsilateral hemisphere as compared to contralateral hemisphere

hsPDA: hemodynamically significant Ductus Arteriosus. PIVH: peri-intraventricular hemorrhages. rScO_2_: regional cerebral oxygen saturation.

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
