# Peer review of "Neuroprotection of the Perinatal Brain by Early Information of Cerebral Oxygenation and Perfusion Patterns"

_ijms, 2021, doi:10.3390/ijms22105389_

Round 1
Reviewer 1 Report
The review entitled “Neuroprotection of the Perinatal Brain by Early Information of Cerebral Oxygenation and Perfusion Patterns” by Costa et al. reports interesting data collection of actual literature about the mechanisms of brain injury of the term infant, cerebral oxygenation and future perspectives of multimodal cerebral oxygenation monitoring. The question addressed in this paper is of current interest given the relevance of this topic. The review is well written, and the sequence of paragraphs is logic.
Author Response
The authors would like to thank this reviewer for the positive feedback on the manuscript.
Reviewer 2 Report
Comments on “neuroprotection of the perinatal brain by early information of cerebral oxygenation and perfusion patterns”
Nowadays, neuronal damage is occurring in a way too high number of pre- and neo-nates, unfortunately. These neuronal damages, mostly related to peri-intraventricular hemorrhages (PIVH) and diffuse white matter injury (dWMI), are associated with hypoxemia, hypo/hypercapnia or cerebral autoregulation dysfunction. Indeed, we need to find tools that can rapidly detect these oxygenation/perfusion abnormalities to be able to reduce or even prevent brain damage. In this review, the authors propose that Near Infrared Spectroscopy (NIRS) as a potential tool that can evaluate oxygen and perfusion levels non-invasively.
Although the topic is very interesting, and the problematic well identified, the authors did not convince this reviewer of the relevance of this review. Even though the advantages of using NIRS are easily identified, the authors did not compare with other methods such as for example functional Molecular Resonance Imaging (fMRI), which is by far the most accurate method to detect oxygenation and perfusion abnormalities. The real interest of using NIRS, to this reviewer’s point of view, is that 1) it’s portative so very easy to transport, 2) less expensive than an MRI and 3) noninvasive. The authors should also have pointed out more thoroughly the disadvantages such as the resolution, the low penetration and the very low precision (5.2%, as mentioned once). Also, PIVH and dWMI are happening far from the brain surface, which makes this reviewer skeptical about how NIRS can be used to identify such pathologies, even though it is likely to detect oxygenation/perfusion abnormality at distance to the site of the brain injury.
Even if the authors respond positively to the following comments, this reviewer highly recommends the authors to consider another journal to publish their findings.
Comments:
- In general, the review is not written in proper English. There are also a decent number of sentences that are 5 lines long, very confusing, which lack of precise vocabulary. This reviewer would advise the authors to have an external English reader who can help with grammar and general form of the manuscript.
- Abstract: globally good, the topic is clearly defined, even though misleading because only focused on NIRS and not the other techniques. Maybe changing the title of the article by adding at the end of the title: “...using NIRS”?
- Introduction:
- The authors should mention the occurrence of perinatal brain damage at the beginning of the manuscript
- The autoregulation should be defined earlier in the text
- The authors mentioned that: “it is widely accepted that normal values of rScO2 in neonates are between 55-85%”, or they only cited themselves... and the range is quite high, so not trustworthy. Please develop more on this aspect.
- Text:
- Please define “extreme preterm neonates”
- Please define “cerebral resistance vessels”
- Is there any more recent study (1980 for this citation) showing that cerebrovascular bed in neonates is more sensitive to pCO2-induced vasodilation compared to adults?
- Please define “hypercarbia”
- Please provide the occurrence of hemodynamically Patent Ductus Arteriosus
- Table 1: The authors wrote “perinatal ischemic arterial stroke” or it is mentioned PAIS in the manuscript. Please correct.
- The Figure1 needs to be adjusted. No scale, no legend concerning the color patterns.
- Figure2: The authors should mention the age of the patient. Hopefully the patient is a neonate, because the heartbeat numbers are concerning.
- The explanation that eliminating extra electrodes potentially reduces injury risks does not make any sense here. Please remove.

Author Response
First of all, the authors would like to thank you for your feedback on our manuscript and the time you took to go through it.
Regarding the specific comments, we have addressed them individually in the document attached. We have taken them into consideration and adapted the manuscript when we felt like the comments were appropriate.
The authors feel, in a way, disappointed on the negative tendency towards our manuscript and the feedback that was given.
It is also noteworthy to mention that the manuscript was part of an invitation to participate in the Special Issue with a contribution on NIRS. Since the authors already expect the audience to have a certain level of expertise in neonatology, we have tried to make this review as concise as possible, without dwelling in topic-specific details.

Reviewer 3 Report
The review submitted focuses on early assessment and guiding abnormal cerebral oxygenation/perfusion patterns to possibly reduce brain injury. The review has been drafted following a comprehensive review of the existing literature in the field. However, there are some minor changes that could be made for better understanding of the readers.
Comments:
- Line 26-27- Why do these methods (stress reduction (preterm neonate), moderate hypothermia (term neonate) and pharmacological (add-on) therapy) work to improve neuroprotection of the perinatal brain specifically? What processes are taken for older patients with cerebral oxygenation and perfusion issues?
- Line 113-typo-'controlled'
- Line 151-153-Tabulate all the various current/existing monitoring methods with their associated preventable outcomes.
- Line 167-Change word from innocent to unassuming or harmless.
- Line 221-Are there any complications to using modern neuroimaging techniques or have certain detection methods provided contradictory or inaccurate information?
- Line 274-275-NIRS-derived intensity signals -What specific applications are there that could be of interest to the reader?
Author Response
Thank you very much for your very helpful comments and questions.
The answers to specific points are in the document attached.

Round 2
Reviewer 2 Report
The authors responded well to the comments.